# Circulatory MicroRNAs in Plasma and Atrial Fibrillation in the General Population: The Rotterdam Study

**DOI:** 10.3390/genes13010011

**Published:** 2021-12-22

**Authors:** Sven Geurts, Michelle M. J. Mens, Maxime M. Bos, M. Arfan Ikram, Mohsen Ghanbari, Maryam Kavousi

**Affiliations:** Department of Epidemiology, Erasmus MC, University Medical Center Rotterdam, 3015 GD Rotterdam, The Netherlands; s.geurts@erasmusmc.nl (S.G.); m.mens@erasmusmc.nl (M.M.J.M.); m.m.bos@erasmusmc.nl (M.M.B.); m.a.ikram@erasmusmc.nl (M.A.I.); m.ghanbari@erasmusmc.nl (M.G.)

**Keywords:** atrial fibrillation, biomarkers, epidemiology, genomics, microRNAs, risk factors, sex-differences

## Abstract

Background: MicroRNAs (miRNAs), small non-coding RNAs regulating gene expression, have been shown to play an important role in cardiovascular disease. However, limited population-based data regarding the relationship between circulatory miRNAs in plasma and atrial fibrillation (AF) exist. Moreover, it remains unclear if the relationship differs by sex. We therefore aimed to determine the (sex-specific) association between plasma circulatory miRNAs and AF at the population level. Methods: Plasma levels of miRNAs were measured using a targeted next-generation sequencing method in 1999 participants from the population-based Rotterdam Study. Logistic regression and Cox proportional hazards models were used to assess the associations of 591 well-expressed miRNAs with the prevalence and incidence of AF. Models were adjusted for cardiovascular risk factors. We further examined the link between predicted target genes of the identified miRNAs. Results: The mean age was 71.7 years (57.1% women), 98 participants (58 men and 40 women) had prevalent AF at baseline. Moreover, 196 participants (96 men and 100 women) developed AF during a median follow-up of 9.0 years. After adjusting for multiple testing, miR-4798-3p was significantly associated with the odds of prevalent AF among men (odds ratio, 95% confidence interval, 0.39, 0.24–0.66, *p*-value = 0.000248). No miRNAs were significantly associated with incident AF. MiR-4798-3p could potentially regulate the expression of a number of AF-related genes, including genes involved in calcium and potassium handling in myocytes, protection of cells against oxidative stress, and cardiac fibrosis. Conclusions: Plasma levels of miR-4798-3p were significantly associated with the odds of prevalent AF among men. Several target genes in relation to AF pathophysiology could potentially be regulated by miR-4798-3p that warrant further investigations in future experimental studies.

## 1. Introduction

Atrial fibrillation (AF) is the most common cardiac arrhythmia worldwide [1,2]. The prevalence of AF is expected to increase steeply in the coming decades due to aging of the population [1,2,3]. Despite the identification of risk factors for AF [4,5,6,7] and improvement in its management, AF still confers a high morbidity and mortality risk [1,2,7]. Furthermore, recent evidence suggests that sex-differences in AF pathophysiology and prognosis exist [8]. Women with AF are older at diagnosis, have a higher prevalence of hypertension, valvular heart disease, and have an increased risk of stroke, myocardial infarction, and mortality in comparison to men [8].

MicroRNAs (miRNAs) are a class of small non-coding RNAs that post-transcriptionally regulate gene expression by complementary binding to target transcripts. Dysregulation of miRNA function could affect pathology of diseases [9]. Extensive studies have also shown the potential of miRNAs to be used as disease biomarkers, as their expression remains stable after drawing blood and they are easily accessible in different types of body fluid [10]. Over the past years, the role of miRNAs in various cardiovascular diseases has received a major interest [11]. MiRNAs have been suggested, among others, as key regulators of electrical remodeling [12], structural remodeling [13], autonomic nerve remodeling [14], calcium handling abnormalities [15], and inflammation [16] of the heart. These functions suggest a role for miRNAs in AF pathophysiology.

Previous studies have identified plasma levels of several miRNAs to be associated with AF [17,18,19,20,21,22,23,24,25]. However, most of these studies were limited to cross-sectional analysis, a subgroup of AF patients, or hypothesis-driven by studying only subsets of specific miRNAs [18]. To date, limited data exist on the association between circulatory miRNAs in plasma and AF in the general population [20,24]. Furthermore, research regarding sex-differences in the associations of miRNAs with AF is sparse. 

In this study, we aimed to investigate the association between circulatory miRNAs in plasma with prevalent and incident AF in the general population using data from the prospective population-based Rotterdam Study to gain more insight into AF pathophysiology. Additionally, we evaluated if the association differs by sex. We further retrieved the predicted target genes of identified miRNAs and examined if any of these target genes have been associated with AF pathophysiology by previous literature. Moreover, we provided an extensive literature review of the previously reported circulatory miRNAs in blood/plasma in association with AF and we provided a detailed review per type of miRNA and the corresponding study characteristics. Subsequently, we did a look up of our findings and compared them with the findings of the association between previously reported circulatory miRNAs with prevalent and incident AF in an attempt to replicate our findings and previous evidence. Finally, we also sought to investigate in silico whether the identified miRNAs are expressed in the heart in an attempt to further unravel the potential underling mechanism that relates the identified miRNAs to AF pathophysiology. 

## 2. Materials and Methods

### 2.1. Study Population

This study was embedded in the Rotterdam Study [26,27]. In short, the Rotterdam Study is a prospective population-based cohort study that investigates the occurrence and progression of risk factors for chronic diseases in middle-age and elderly persons. The design of the Rotterdam Study is explained in detail in the Appendix A [26,27].

For the present study, we included 1000 participants from the fourth visit of RS-I (RS-I-4) and 1000 participants from the second visit of RS-II (RS-II-2) for whom miRNA expression data were obtained (n = 2000). These visits took place between 2002 and 2005 and we considered this as the baseline of our study. From these 2000 randomly chosen participants, one participant was excluded, due to insufficient baseline data on AF for the cross-sectional study with prevalent AF. For the longitudinal study with incident AF, we additionally excluded prevalent AF cases (n = 98). A total of 1999 participants were included in the current study.

### 2.2. Assessment of Circulatory microRNAs in Plasma

Methods on plasma miRNA level measurement have been described previously [28]. In short, the expression levels of 2083 mature human miRNAs (HTG Molecular Diagnostics, Tuscon, AZ, USA) and using the Illumina NextSeq 500 sequencer (Illumina, San Diego, CA, USA) were measured. Out of 2083 measured miRNAs, 591 miRNAs were well-expressed in plasma [28]. The assessment of miRNAs is further explained in the Appendix A.

### 2.3. Assessment of Atrial Fibrillation

AF was defined in accordance with the European Society of Cardiology (ESC) guidelines [7]. Methods on event adjudication for prevalent and incident AF have been described previously [3]. In short, a 10 s 12-lead electrocardiogram (ECG) was used to assess AF at baseline and additional follow-up information was obtained from medical files and follow-up examinations at the research center. All participants were followed from the date of enrollment in the Rotterdam Study until the date of onset of AF, date of death, loss to follow-up, or to 1 January 2014, whichever occurred first. The assessment of AF is further explained in detail in the Appendix A [3,26,29].

### 2.4. Assessment of Cardiovascular Risk Factors

The cardiovascular risk factors included in the study were body mass index (BMI), total cholesterol, high-density lipoprotein (HDL) cholesterol, hypertension, smoking status, history of diabetes mellitus (DM), history of coronary heart disease (CHD), history of heart failure (HF), left ventricular hypertrophy (LVH) on the ECG, use of cardiac medication, and use of lipid lowering medication. Methods for measurements of cardiovascular risk factors are explained in detail in the Appendix A [26,27].

### 2.5. Statistical Analyses

Participant characteristics at study entry are presented as mean with standard deviation (SD) or number (n) with percentages as appropriate. Group differences between men and women were examined by Student’s *t*-test for continuous variables and Chi-Square Test for categorical variables.

Logistic regression and Cox proportional hazards models were used to assess the association between plasma miRNAs at baseline with prevalent and incident AF, respectively (Figure 1). Odds Ratios (ORs) and hazard ratios (HRs) with 95% confidence intervals (CIs) were calculated to quantify the associations. An examination of the shape of relation with AF was performed using natural cubic splines for continuous variables, and no deviation from linearity was found. No influential values were observed when using Cook’s distance and no multicollinearity among the variables was observed using a variance inflation factors threshold of <5. The proportional hazards assumptions were tested by Schoenfeld tests and were found to be satisfied. Additionally, we examined the interaction of miRNAs and sex before subsequently stratifying our analyses.

Analyses were performed in the total study population and for men and women separately. All models were adjusted for age, sex (if applicable), and cohort (model 1) and additionally for cardiovascular risk factors including BMI, total cholesterol, HDL cholesterol, hypertension, smoking status, history of DM, history of CHD, history of HF, LVH on the ECG, use of cardiac medication, and use of lipid lowering medication (model 2). Missing values of variables were imputed under the assumption of missing at random using multiple imputation. For multiple imputation, all available data were used to generate an imputed dataset. 

The *p*-value threshold was corrected for multiple testing based on the eigenvalues of the correlation matrix from all the miRNAs. This adapted method was proposed by Li [30] and is based on a method which was introduced by Cheverud [31] to adjust correlated tests as if they were independent, according to an “effective number” of independent tests, as there is evidence that miRNAs are clustered together or may be co-expressed [32]. This means that the miRNAs are thereby correlated with each other and by adopting this *p*-value correction proposed by Li [30] we take this correlation into account when adjusting. Based on the aforementioned method from Li, [30] the significance *p*-value cutoff was set at 0.000352 based on 142 identified independent tests (0.05/142). 

The data management and analyses were performed using IBM SPSS Statistics version 25.0 for Windows (IBM Corp., Armonk, NY, USA) and R software (R 4.0.2; R Foundation for Statistical Computing, Vienna, Austria) [33].

### 2.6. Assessment of Predictive Target Genes

We retrieved a list of predicted targets genes of identified miRNAs associated with AF using the three commonly used target prediction databases: miRDB [34,35], TargetScan [36], and miRTarBase [37]. Furthermore, we assessed if any of these predicted target genes have been associated previously with AF by a systematic review and a genome-wide association study [18,38] and we assessed if any of these genes are potentially involved in AF pathophysiology by electrical and/or structural remodeling of the heart (Figure 1).

### 2.7. Literature Review

We searched the literature (PubMed) to identify studies that reported on circulatory miRNAs in blood/plasma in association with AF. Subsequently, we tested the association of these previously reported circulatory miRNAs with prevalent and incident AF in our study in an attempt to replicate our findings and previous findings (Figure 1).

### 2.8. In Silico Analyses

We also sought to investigate whether the identified miRNAs are expressed in the heart in an attempt to further understand the potential underlying mechanism that might link the identified miRNAs to AF pathophysiology. In addition, we retrieved the miRNA host genes as proxy for the identified miRNAs to evaluate their expression in the heart using the Human Protein Atlas (Figure 1) [39]. The idea behind this is that intragenic miRNAs and their host genes are likely to be co-expressed [40]. Furthermore, the genomic location of the identified miRNAs was obtained using miRIAD [41].

## 3. Results

### 3.1. Statistical Analyses

A total of 1999 participants, 858 men (42.9%) and 1141 women (57.1%), were eligible for the analyses of miRNAs associated with prevalent AF. The baseline characteristics for the study sample are depicted in Table 1. For the longitudinal analyses of incident AF, after exclusion of prevalent AF cases, 1901 individuals, 800 men (42.1%) and 1101 women (57.9%), were included. Characteristics of this study population are presented in Appendix A. Compared to men, women were slightly older, more often hypertensive and never smokers. DM, CHD, HF, and LVH on the ECG were less prevalent among women. Women used cardiac medication and lipid lowering medication less frequently than men.

At baseline, 98 cases (4.9%) of prevalent AF were identified, from which 58 cases (6.8%) were in men and 40 cases (3.5%) were in women. Logistic regression showed that 47 miRNAs in the total study population, 45 miRNAs in men, and 31 miRNAs in women were nominally significantly (*p*-value < 0.05) associated with prevalent AF after adjustment for age and cardiovascular risk factors (model 2). See Appendix A for an overview of the nominally significantly associated miRNAs. For one unit increase in miR-4798-3p plasma levels at baseline, the odds for prevalent AF in the total study population was OR, 95% CI, 0.64, 0.44–0.97, *p*-value = 0.028033 (model 1). After adjusting for cardiovascular risk factors, the odds did not attenuate OR, 95% CI, 0.63, 0.42–0.99, *p*-value = 0.034433 (model 2). The odds for prevalent AF were lower in men than in women. After adjustment for cardiovascular risk factors, ORs, 95% CIs were 0.39, 0.24–0.66, *p*-value = 0.000248 in men and 1.84, 0.76–4.97, *p*-value = 0.203587 in women (model 2). However, after adjusting for multiple testing (0.05/142 = 0.000352), only miR-4798-3p, remained statistically significantly associated with prevalent AF among men (see Table 2 for more details). The interaction term between miRNA-4798-3p and sex in relation to the odds of prevalent AF in the total study population using logistic regression was significant (*p*-value = 0.004730). This significant sex interaction further highlights our observed sex-differences for miR-4798-3p. Figure 2 illustrates the nominally significant miRNAs associated with prevalent AF among men described by a volcano plot. 

During a median follow-up of 9.0 years (interquartile range (IQR), 7.7–10.3), 196 incident AF cases (10.3%) (96 in men and 100 in women) occurred. The incidence rate of AF was 12.5 per 1000 person-years in the total study population (15.2 per 1000 person years in men, 10.7 per 1000 person years in women). 

Cox proportional hazards models showed that a total of 17 miRNAs in the total study population, 26 miRNAs in men, and 13 miRNAs in women were nominally significant in association with incident AF (model 2), but none of them remained statistically significant after adjustment for multiple testing. Appendix A shows a complete list of the nominally significantly associated miRNAs with incident AF.

There was little overlap in similarity between the effect estimates of the miRNAs among the prevalent AF cases when we compared them to the effect estimates in the incident AF sample and vice versa. This was also the case when we compared the effect estimates of miR-4798-3p for the association with prevalent AF among men with the effect estimates of miR-4798-3p for incident AF among men (OR, 95% CI, 0.39, 0.24–0.66, *p*-value = 0.000248 vs. HR, 95% CI, 1.10, 0.68–1.77, *p*-value = 0.704529) (model 2).

### 3.2. Predictive Target Genes

We additionally examined the predicted target genes of miR-4798-3p using three miRNA target prediction databases: miRDB [34,35], TargetScan [36], and miRTarBase [37]. To reduce error, we only retained predicted target genes if they were identified by at least two out of the three databases that were used [42]. Among predicted target genes of miR-4798-3p are *CACNB2* [43], *KCNN3* [44], *SIRT1* [45], and *STAT3* [46,47] that are suggested to be involved in electrical and/or structural remodeling of the heart. Appendix A depicts the genes, that were among the predicted target genes of miR-4798-3p, and that have been previously associated with AF by a systematic review and a genome-wide association study [18,38]. In addition, the potential remodeling mechanisms of the heart for these genes are also provided [48]. 

### 3.3. Literature Review

Additionally, we provided an extensive literature review of circulatory miRNAs in blood/plasma in association with AF that have been reported in the literature before (Appendix A). In this review, we provided detailed information per type of miRNA and the corresponding study characteristics including study design, study population, baseline characteristics, reported effect estimates, the statistical models, and adjustments. Furthermore, we did a look-up for these AF-associated miRNAs in our results in an attempt to replicate our results and previous results. The effect estimates and *p*-values for the association of these previously reported miRNAs with prevalent and incident AF in our data are reported in Appendix A, respectively. For the prevalent AF analyses, we were able to compare 39 miRNAs. Among these 39 miRNAs, the direction of the effect estimates reported in the literature were in line with our results for 18 miRNAs. The reported effect estimate in the literature that was most similar to our findings was the effect estimate for miR-20a-5p (literature-reported OR, 95% CI, 1.36, 1.14–1.61, *p*-value = 0.001 [24], while we found an OR, 95% CI, 1.30, 0.68–2.58, *p*-value = 0.435846). Moreover, for the incident AF analyses, we were able to compare 10 miRNAs from the literature with our results. The direction of the effect was similar for 4 miRNAs and the miRNA with the most similar effect estimate was miR-193a-5p with a literature-reported HR, 95% CI, 0.87, 0.77–0.98, *p*-value = 0.024 [20], while we found a HR, 95% CI, 0.93, 0.67–1.28, *p*-value = 0.640964.

### 3.4. In Silico Analyses

Finally, we explored whether miR-4798-3p was expressed in the heart. We were unable to find any information regarding its expression levels within the heart [39]. Alternatively, we evaluated the expression of its host gene as a proxy for miR-4798-3p [40]. MiR-4798-3p is located within an intron of the protein-coding gene *SORCS2* [41]. *SORCS2* is especially found within the central nervous system, and it is well-expressed within the brain and to a lesser degree within the heart [39]. Moreover, the expression levels of AF-associated target genes of miR-4798-3p (Appendix A) were also detected in various degrees within the heart [39].

## 4. Discussion

In this prospective population-based study, we conducted a systematic analysis of 591 circulatory miRNAs well-expressed in plasma with the odds and the risk of AF in the general population and for men and women separately. We found that plasma levels of miR-4798-3p were significantly associated with the odds of prevalent AF among men after extensive adjustment for potential confounders and correcting for multiple testing. Several predicted target genes of miR-4798-3p have been associated previously with AF in a systematic review [18] and data from a recent genome-wide association study on AF [38]. These target genes are potentially involved in electrical and/or structural remodeling of the heart and thereby may mediate the effect of miR-4798-3p in AF pathophysiology. Future experimental studies are warranted to investigate the potential (sex-specific) role of this miRNA in molecular pathways underlying AF. We also provided an extensive and detailed literature review of the previously reported miRNAs linked to AF and compared these literature-reported associations with the associations observed in our study in attempt to replicate our results and previous studies. Lastly, we investigated in silico if miRNA-4798-3p and its AF-associated target genes are expressed within heart.

Plasma levels of miR-4798-3p were significantly associated with the odds of prevalent AF in our study. The exact pathology behind the associations of many miRNAs with cardiovascular diseases is not completely understood. In general, miRNAs are involved in every biological pathway through regulating expression of target genes/transcripts. However, no other studies have identified circulatory miR-4798-3p in plasma as a potential risk factor or biomarker for atrial fibrillation. MiR-4798-3p has a predicted number of more than 50 target genes that it may regulate [34,35,36,37]. Various genes which are potentially regulated by this miRNA are involved in calcium and potassium handling in myocytes (*CACNB2*, *KCNN3*), in protection of cells against oxidative stress (*SIRT1*), and in regulating cardiac fibrosis (*STAT3*). These aforementioned mechanisms are linked to electrical and/or structural remodeling of the heart which are associated with AF pathophysiology [8,12,18,38,49]. The host gene *SORCS2* is profoundly expressed within the central nervous system and may thereby potentially exert an effect on AF vulnerability, as an effect on the extensive network of vagal ganglionated plexi is known to affect AF risk [50]. However, if these circulating levels of miRNA in plasma by themselves cause AF, or if the circulating levels of miRNAs are merely a reflection of an underlying pathology that may lead to the pathogenesis of AF, is not clear. In addition, it is beyond the scope of this investigation to elucidate on the pathophysiologic implications of a host gene and putative target genes. Future experimental studies are warranted to investigate the interaction between miR-4798-3p, its host gene, its target genes, and their relation to AF. These future experimental studies could then further aid in early AF diagnosis, risk stratification, therapeutic monitoring, or identification of potential interesting pharmaceutical drug targets among men. The limited overlap between miRNAs associated with prevalent and incident AF might suggest that miRNAs may indeed be a reflection of underlying pathology that is associated with prevalent AF instead of that miRNAs may cause incident AF over time. The potential discrepancy between cell-specific expression of miRNAs and circulatory (cell-free) miRNAs in plasma makes it more difficult to disentangle this pathophysiology. However, it has been shown that circulatory miRNAs constitute a way of cell-to-cell communication, and miRNAs are released to extracellular matrix and blood by exosome from the diseased tissue/cells. Moreover, the duration of time that is involved in the release of miRNA in plasma (by a pathological event), and the effect that it may have, is still elusive. Nevertheless, miR-4798-3p could still be a potentially useful plasma biomarker for AF prediction or prognosis.

Sex-differences in AF pathophysiology are increasingly gaining interest [8].The association of miR-4798-3p with prevalent AF in our study was only significant among men. This difference could be explained by the different target genes of miR-4798-3p and their potential sex-specific effects. For example, *KCNN3* and *CACNB2* regulate L-type calcium channels and may thereby influence QT intervals of the heart [51]. Women have different and longer QT-intervals than men [8], and a long QT-interval has been associated with AF initiation [52]. *SIRT1* is known to be upregulated in patients with CHD, possibly as a potential compensatory mechanism to counteract the adverse effects of oxidative stress caused by CHD [53,54]. CHD is more prevalent among men than in women [8] and is implicated in AF pathophysiology [4,5,6,7,8]. *STAT3* is involved in cardiac fibrosis and previous research has shown that women with AF have more atrial fibrosis than men [8]. Although these effects could be sex-specific, further exploration is warranted to examine the exact underlying molecular mechanisms that might explain these sex-differences. 

To the best of our knowledge, the Framingham Heart Study is the only population-based cohort study that has previously investigated the association between miRNAs and AF at the population level. McManus et al. [20] identified one miRNA that was significantly associated with prevalent AF (miR-328) in the Framingham Heart Study, while they did not find any significant miRNAs that were associated with incident AF. Vaze et al. [24] identified six miRNAs that were significantly associated with incident AF in the Framingham Heart Study, including four also significantly associated with prevalent AF (miR-106b, miR-26a-5p, miR-484, and miR-20a-5p). We could not replicate our findings and the findings from McManus et al. [20] and Vaze et al. [24], or the results from the studies assessed during the literature review. These differences may be due to the fact that we measured circulatory miRNA levels in plasma instead of whole blood [55] as in the Framingham Heart Study, differences in miRNA expression profiling [56], differences in adjusting for confounders, differences in correcting for multiple testing, and differential expression of miRNAs related to the type and phase of AF. It is worth noting that an internationally adopted standardized method to evaluate miRNA expression could potentially improve future miRNA studies (for example plasma vs. blood or circulatory vs. tissue). As such, a standardization could then improve the comparability between future miRNA studies, and this would also benefit any potential clinical applications of miRNA-based therapies in the future. However, our findings extend the aforementioned studies by examining 591 (instead of 253–339) miRNAs, a longer follow-up time, and more extensive adjustment for potential confounding. Additionally, we also examined potential sex-differences in the associations between miRNAs and AF in our study population, and we thereby also add to the emerging evidence that circulating miRNAs play a critical role in the pathophysiology of AF that may potentially be sex-specific. 

Major strengths of our study include its population-based nature, large sample size, precise adjudication of prevalent and incident AF, detailed information on cardiovascular risk factors, a long follow-up time, including a well-expressed set of 591 miRNAs, extensive adjustment for potential confounders, the examination of potential sex-differences between miRNAs and AF, our detailed literature review, and in silico analyses to further understand the potential underlying mechanisms. Nonetheless, there are some limitations. We could not distinguish between paroxysmal, persistent, long-standing persistent, and permanent AF, as Holter monitoring is not available in this large population-based cohort study. Although, we extensively adjusted for confounders, residual confounding cannot be entirely ruled out. Furthermore, our study population included mainly elderly participants from European descent and our results may therefore not be generalizable to younger populations or other ethnicities. 

In this large population-based cohort study we assessed 591 well-expressed circulatory miRNAs in plasma in relation to prevalent and incident AF. We found that plasma levels of miR-4798-3p were significantly associated with the odds of prevalent AF among men. Several target genes in relation to AF pathophysiology could potentially be regulated by miR-4798-3p that warrant further investigations in future experimental studies. 

## Figures and Tables

**Figure 1 genes-13-00011-f001:**
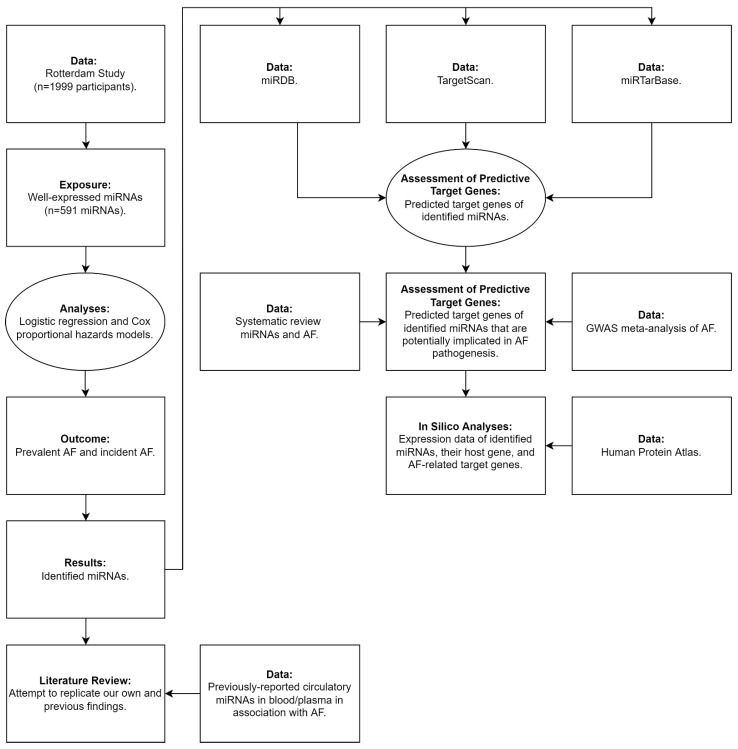
Flow chart for the conducted analyses and search strategy. **Abbreviations:** AF, atrial fibrillation; GWAS, genome-wide association study; miRNAs, microRNAs.

**Figure 2 genes-13-00011-f002:**
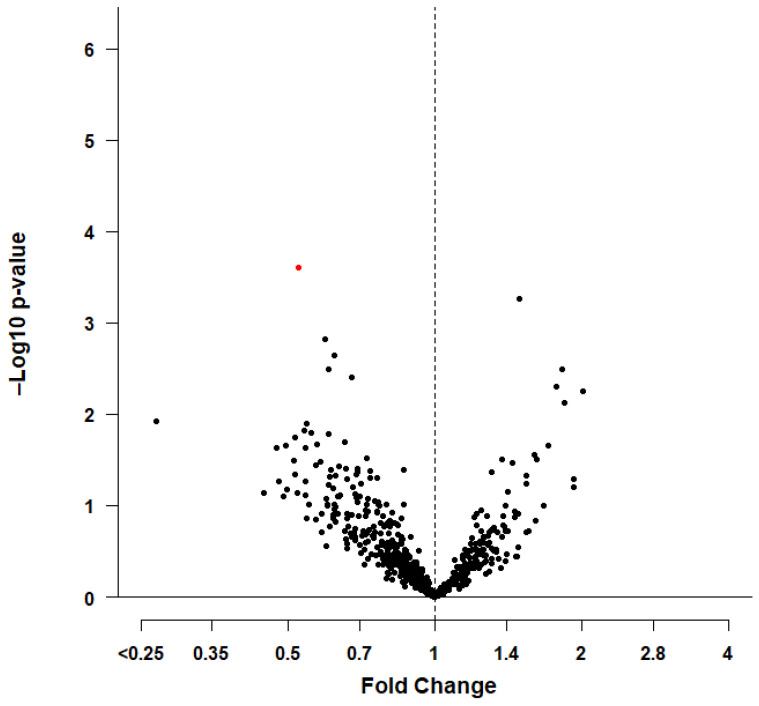
Volcano plot of nominally significant miRNAs in association with prevalent atrial fibrillation among men. The y-axis of Figure 2 represents the negative log of the p-value on the y-axis and the x-axis represents the log of the fold change for prevalent AF. The black dots indicate the associations of the nominally significant miRNAs. The red dot indicates the significant association after correction for multiple testing of miR-4798-3p.

**Table 1 genes-13-00011-t001:** Baseline characteristics of the total study population and stratified by sex.

Baseline Characteristics *	Total Study Population n = 1999	Men n = 858	Women n = 1141	*p*-Value †
Age, years	71.7 ± 7.6	71.4 ± 7.3	71.9 ± 7.8	0.116
Women, n (%)	1141 (57.1)	NA	1141 (100)	
Body mass index, kg/m^2^	27.7 ± 4.1	27.6 ± 3.4	27.7 ± 4.6	0.382
Total cholesterol, mmol/L ‡	5.6 ± 1.0	5.3 ± 1.0	5.9 ± 1.0	<0.001
High-density lipoprotein cholesterol, mmol/L ‡	1.4 ± 0.4	1.0 ± 0.3	1.6 ± 0.4	<0.001
Hypertension, n (%)	1558 (77.9)	654 (76.2)	904 (79.2)	0.109
Smoking status, n (%)				<0.001
Never	599 (30.0)	117 (13.6)	482 (42.2)	
Former	1094 (54.7)	592 (69.0)	502 (44.0)	
Current	306 (15.3)	149 (17.4)	157 (13.8)	
History of diabetes mellitus, n (%)	268 (13.4)	145 (16.9)	123 (10.8)	<0.001
History of coronary heart disease, n (%)	213 (10.7)	145 (16.9)	68 (6.0)	<0.001
History of heart failure, n (%)	101 (5.1)	50 (5.8)	51 (4.5)	0.170
Left ventricular hypertrophy, n (%)	108 (5.4)	62 (7.2)	46 (4.0)	0.002
Cardiac medication, n (%)	210 (10.5)	102 (11.9)	108 (9.5)	0.108
Lipid lowering medication, n (%)	450 (22.5)	208 (24.2)	242 (21.2)	0.080

Abbreviations: n, number; NA, not applicable. The table presents baseline characteristics of the total study population for the analyses of prevalent atrial fibrillation. * Values are mean (standard deviation) or number (percentages). † Statistical significance between men and women for continuous data was tested using the Student’s *t*-test and for categorical data was tested using the Chi-Square Test. ‡ SI conversion factors: to convert cholesterol to mg/dL divide by 0.0259.

**Table 2 genes-13-00011-t002:** Association between miR-4798-3p with the odds of prevalent atrial fibrillation in the total study population and stratified by sex.

	OR (95% CI)			
	Model 1 *	*p*-Value	Model 2 †	*p*-Value
**Total Study Population**				
**miR-4798-3p**	0.64 (0.44–0.97)	0.028033	0.63 (0.42–0.99)	0.034433
**Men**				
**miR-4798-3p**	0.42 (0.27–0.69)	0.000254	0.39 (0.24–0.66)	0.000248
**Women**				
**miR-4798-3p**	1.53 (0.71–3.70)	0.311964	1.84 (0.76–4.97)	0.203587

Abbreviations: CI, confidence interval; OR, odds ratio. * Adjusted for age, and cohort (model 1). † Adjusted for age, cohort, body mass index, total cholesterol, high-density lipoprotein cholesterol, hypertension, smoking status, history of diabetes, history of coronary heart disease, history of heart failure, left ventricular hypertrophy on the electrocardiogram, use of cardiac medication, and use of lipid lowering medication (model 2).

## Data Availability

Data can be obtained upon request. Requests should be directed towards the management team of the Rotterdam Study (secretariat.epi@erasmusmc.nl), which has a protocol for approving data requests. Due to restrictions based on privacy regulations and informed consent of the participants, data cannot be made freely available in a public repository.

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
