# Peer review of "Circulatory MicroRNAs in Plasma and Atrial Fibrillation in the General Population: The Rotterdam Study"

_genes, 2021, doi:10.3390/genes13010011_

Round 1

Reviewer 1 Report

This is an original research article, which aims to investigate the association between circulatory miRNAs in plasma with prevalent and incident of AF in the general population using data from a prospective population-based cohort study “Rotterdam Study”, to examine possible sex-differences between miRNAs and AF, to provide an extensive literature overview of the previously reported circulatory miRNAs in blood or plasma linked to AF and to compare them with the associations arrived from this study, as well to evaluate in silico miRNAs expression in heart, using Human Protein Atlas.

Generally, the authors have in depth knowledge of this topic, and they have used the appropriate methodology, study design, and well-performed statistical analysis. The results are sufficiently well-presented, clear and easy to understand, so as to reach safe and solid conclusions. Overall, the manuscript is well written and structured, and all sections are well-developed. Thus, I think it would make a nice addition to Genes as an original research article. However, the following points should be considered.

1)This study has some limitations with regard to the AF type (permanent or paroxysmal) and limited patient characteristics, as referred in the manuscript.

2)Grammatical, syntax, typographical, as well punctuation errors were noted through the manuscript. All these need to be identified and also to be corrected.

I also suggest the following minor modifications at specific parts of this manuscript.

1)Please refer any data with regard to LDL cholesterol levels, family history of CVD, or duration of CVD, type of cardiac medication, if available, in the part “assessment of cardiovascular risk factors” and further present them in table 1.

2)Apply the same format to the whole supplement. I suggest you justify the text so as to look neater.

3)Present most recent and further data with regard to miR-4798-3p either in the part of introduction or discussion.

Author Response

Dear editors and reviewers,

We highly appreciate your thoughtful consideration of our manuscript and would like to thank the reviewers for their valuable comments that have been very useful in improving the manuscript. We have addressed the comments to the best of our ability. We believe that the manuscript is stronger, as the result of thoughtful comments of the reviewers, and hope that it now merits publication in Genes.

Below, please see our point-by-point responses to the reviewers’ comments.

Reviewer #1
This is an original research article, which aims to investigate the association between circulatory miRNAs in plasma with prevalent and incident of AF in the general population using data from a prospective population-based cohort study “Rotterdam Study”, to examine possible sex-differences between miRNAs and AF, to provide an extensive literature overview of the previously reported circulatory miRNAs in blood or plasma linked to AF and to compare them with the associations arrived from this study, as well to evaluate in silico miRNAs expression in the heart, using Human Protein Atlas.

Generally, the authors have in depth knowledge of this topic, and they have used the appropriate methodology, study design, and well-performed statistical analysis. The results are sufficiently well-presented, clear and easy to understand, so as to reach safe and solid conclusions. Overall, the manuscript is well written and structured, and all sections are well-developed. Thus, I think it would make a nice addition to Genes as an original research article. However, the following points should be considered.
Response of the author:
We thank the reviewer for the compliments regarding our manuscript and acknowledging the importance, novelty, and strengths of our study.

Comments:
1. This study has some limitations with regard to the AF type (permanent or paroxysmal) and limited patient characteristics, as referred in the manuscript.

Response of the author: This is the limitation pertinent to the prospective population-based cohort studies ascertaining AF through follow-up visits and records. We have added additional information as follows: “We could not distinguish between paroxysmal, persistent, long-standing persistent, and permanent AF as Holter monitoring is not available in this cohort” in the limitations of our study in Discussion on page 11, lines 397-398.

2. Grammatical, syntax, typographical, as well punctuation errors were noted through the manuscript. All these need to be identified and also to be corrected.
Response of the author:
We have corrected all such instances throughout the manuscript where applicable.

I also suggest the following minor modifications at specific parts of this manuscript.

3. Please refer any data with regard to LDL cholesterol levels, family history of CVD, or duration of CVD, type of cardiac medication, if available, in the part “assessment of cardiovascular risk factors” and further present them in Table 1.
Response of the author:
We thank the reviewer for this comment. However, some of this information is unfortunately not available in this study cohort (RS-I-4 and RS-II-2). With regards to type of medication. Medication use was derived from baseline questionnaires, pharmacy data, and was categorized and defined according to the World Health Organization Anatomical Therapeutic Chemical (WHO ATC) classification. We added additional information “More specifically, cardiac medication, antihypertensive medication, and lipid lowering medication were defined according to the WHO ATC categories c01, c02, and c10 respectively.” in the Supplementary Material on page 5-6, lines 112-114.

4. Apply the same format to the whole supplement. I suggest you justify the text so as to look neater.
Response of the author:
We applied the same format to the text where applicable.

6. Present most recent and further data with regard to miR-4798-3p either in the part of introduction or discussion.
Response of the author:
Following the advice from the reviewer, we have added additional information: “However, no other studies have identified circulatory miR-4798-3p in plasma as a potential risk factor or biomarker for atrial fibrillation.” in the Discussion on page 10, lines 324-326.

Reviewer 2 Report

This is an interesting paper, and in this manuscript, Seven Geurts et al. have studied the circulatory microRNAs in the plasma of AF patients from the population in Rotterdam and established an association between a novel miRNA and their target genes in AF.

Though, previous studies have identified several miRNAs to be associated with AF in the plasma. However, most of these studies were limited to cross-sectional analysis, a 60 subgroup of AF patients, or hypothesis-driven by studying only subsets of specific miR-61 NAs. In this manuscript, the authors have found a convincing link between plasma miR-4798-3p level and prevalent AF among men. They also identified several target genes of the miR-35-4798 in relation to AF pathophysiology.

Additionally, the research is also focused on sex differences in the associations of miRNAs with AF. This research is also limited, and this study tends to gain more insight into AF pathophysiology. The study overall is well organized and executed soundly. All the statistical parameters were well defined. The manuscript displayed novel research and is of significance. Hence it is suitable for the journal.

Addressing the following comments would undoubtedly improve the manuscript.

 Comments :

  1. The in silico identified miRNAs and their targets could be validated in order to confirm the expression of the miRNAs in the heart. For instance, the orthologue of the gene can be studied in animal models such as zebrafish, mice, etc. The expression of this gene can be confirmed by various methods such as classical in situ hybridization or advanced method RNAscope.
  2. Did the authors adopt any precautions to reduce error? For example, miRWalk 2.0 can be applied for miRNA target prediction, which composed of miRDB, TargetScan, and miRTarBase and to reduce error, the final identification of miRNA target genes was in line with at least two of the above tools as reported in Shen NN, Zhang C, Li Z, et al. MicroRNA expression signatures of atrial fibrillation: The critical systematic review and bioinformatics analysis. Exp Biol Med (Maywood). 2020;245(1):42-53. doi:10.1177/1535370219890303.
  3. Based on the identified target genes, a network map of miRNA target genes could be performed using tools such as Cytoscape.  Cytoscape could be applied to visualize molecular interaction networks between miRNAs and corresponding gene targets. 
  4. The authors may conduct biological analysis of KEGG (Kyoto Encyclopedia of Genes and Genomes) and GO (Gene Ontology) analysis according to the predicted targets, and the significance of the associated biological analysis can be confirmed with the evaluation of the DAVID database. This will help in identifying enrichment of the gene targeted by miRNAs.
  5. Some of the crucial references are missing. For instance, “Sumra Komal, Jian-Jian Yin, Shu-Hui Wang, Chen-Zheng Huang, Hai-Long Tao, Jian-Zeng Dong, Sheng-Na Han, Li-Rong Zhang, MicroRNAs: Emerging biomarkers for atrial fibrillation, Journal of Cardiology is missing in the manuscript. Again Shen NN, Zhang C, Li Z, et al. MicroRNA expression signatures of atrial fibrillation: The critical systematic review and bioinformatics analysis. Exp Biol Med (Maywood). 2020;245(1):42-53. doi:10.1177/1535370219890303 The authors may consider them as they are relevant studies.
  6. More importantly, a flowchart for the search strategy in this study or a bioinformatic analysis pipeline would be very helpful in understanding the work. The authors may consider incorporating a flowchart.

Author Response

Dear editors and reviewers,

We highly appreciate your thoughtful consideration of our manuscript and would like to thank the reviewers for their valuable comments that have been very useful in improving the manuscript. We have addressed the comments to the best of our ability. We believe that the manuscript is stronger, as the result of thoughtful comments of the reviewers, and hope that it now merits publication in Genes.

Below, please see our point-by-point responses to the reviewers’ comments.

Reviewer #2
This is an interesting paper, and in this manuscript, Sven Geurts et al. have studied the circulatory microRNAs in the plasma of AF patients from the population in Rotterdam and established an association between a novel miRNA and their target genes in AF.

Though, previous studies have identified several miRNAs to be associated with AF in the plasma. However, most of these studies were limited to cross-sectional analysis, a 60 subgroup of AF patients, or hypothesis-driven by studying only subsets of specific miR-61 NAs. In this manuscript, the authors have found a convincing link between plasma miR-4798-3p level and prevalent AF among men. They also identified several target genes of the miR-4798-3p in relation to AF pathophysiology.

Additionally, the research is also focused on sex-differences in the associations of miRNAs with AF. This research is also limited, and this study tends to gain more insight into AF pathophysiology. The study overall is well organized and executed soundly. All the statistical parameters were well defined. The manuscript displayed novel research and is of significance. Hence it is suitable for the journal.
Response of the author:
We thank the reviewer for the recognizing the importance, novelty, and strengths of our study. We have addressed the comments of the reviewer in our point-by-point response.

Addressing the following comments would undoubtedly improve the manuscript.

Comments :
1. The in silico identified miRNAs and their targets could be validated in order to confirm the expression of the miRNAs in the heart. For instance, the orthologue of the gene can be studied in animal models such as zebrafish, mice, etc. The expression of this gene can be confirmed by various methods such as classical in situ hybridization or advanced method RNAscope.
Response of the author:
We acknowledge the reviewer’s comment. Indeed, experimental validation studies in vitro are needed to further investigate the underlying molecular mechanism and confirm our findings, however it is beyond the scope of the current manuscript. This could be something for future studies to validate the miRNA host gene, and putative target genes and to elucidate on the pathophysiologic implications of this host gene and putative target genes in cardiovascular diseases. We have conducted our analyses on a population-based level (at our department of Epidemiology) and we do not have access to lab facilities for the follow-up experimental studies.

2. Did the authors adopt any precautions to reduce error? For example, miRWalk 2.0 can be applied for miRNA target prediction, which composed of miRDB, TargetScan, and miRTarBase and to reduce error, the final identification of miRNA target genes was in line with at least two of the above tools as reported in Shen NN, Zhang C, Li Z, et al. MicroRNA expression signatures of atrial fibrillation: The critical systematic review and bioinformatics analysis. Exp Biol Med (Maywood). 2020;245(1):42-53. doi:10.1177/1535370219890303.
Response of the author:
We thank the reviewer for this comment. We updated our final identification method to be in line with Shen NN, Zhang C, Li Z, et al. miRDB and TargetScan. Additionally, we added this additional information “To reduce error, we only retained predicted target genes if they were identified by at least two out of the three both databases that were used” on page 4, lines 161-163 and on page 8, lines 266-269.

3. Based on the identified target genes, a network map of miRNA target genes could be performed using tools such as Cytoscape. Cytoscape could be applied to visualize molecular interaction networks between miRNAs and corresponding gene targets.
Response of the author:
Considering the amount of potential target genes of miR-4798-3p that were identified by the three databases that we used (range 50-1543). We would not be in favour to visualize these as the network would be too big for any meaningful interpretation. Further, this figure would then also include putative target genes that may not be relevant for the potential underlying biology. This indeed would be a good suggestion after performing experimental validation studies and make a short list of validated target genes.

4. The authors may conduct biological analysis of KEGG (Kyoto Encyclopedia of Genes and Genomes) and GO (Gene Ontology) analysis according to the predicted targets, and the significance of the associated biological analysis can be confirmed with the evaluation of the DAVID database. This will help in identifying enrichment of the gene targeted by miRNAs.
Response of the author:
Although, we did not conduct a formal biological analysis using KECGG or GO. We did identify the most relevant predicted target genes of miR-47983-3p that have been previously associated with atrial fibrillation and assessed their potential remodeling mechanism in Supplementary Table 4. This table potentially explains on how these target genes are implicated in AF pathogenesis, so we already did provide an explanation of our findings for putative target genes and its potential underlying biology.

5. Some of the crucial references are missing. For instance, “Sumra Komal, Jian-Jian Yin, Shu-Hui Wang, Chen-Zheng Huang, Hai-Long Tao, Jian-Zeng Dong, Sheng-Na Han, Li-Rong Zhang, MicroRNAs: Emerging biomarkers for atrial fibrillation, Journal of Cardiology is missing in the manuscript. Again Shen NN, Zhang C, Li Z, et al. MicroRNA expression signatures of atrial fibrillation: The critical systematic review and bioinformatics analysis. Exp Biol Med (Maywood). 2020;245(1):42-53. doi:10.1177/1535370219890303 The authors may consider them as they are relevant studies.
Response of the author:
Komal et al. was already among our references and we added Shen et al. to our references.

6. More importantly, a flowchart for the search strategy in this study or a bioinformatic analysis pipeline would be very helpful in understanding the work. The authors may consider incorporating a flowchart.
Response of the author:
We thank the reviewer for this suggestion. We have added a new figure: Figure 1 on page 5 as requested. Moreover, additional information in the Methods on page 3, line 132; page 4, lines 167, 172, and 178 was added.

Author Response

Dear editors and reviewers,

We highly appreciate your thoughtful consideration of our manuscript and would like to thank the reviewers for their valuable comments that have been very useful in improving the manuscript. We have addressed the comments to the best of our ability. We believe that the manuscript is stronger, as the result of thoughtful comments of the reviewers, and hope that it now merits publication in Genes.

Below, please see our point-by-point responses to the reviewers’ comments.

Reviewer #3
These authors have measured circulatory miRNAs in plasma as it is associated with prevalent atrial fibrillation.
The large number of subjects studied (~2,000), the large number of miRNAs (591) studied, and the long follow-up are strengths of this work.

1. The author should consider listing miR-4798-3p in the title of the paper. If this marker becomes important for routine clinical use, it will be discovered and cited.
Response of the author:
We acknowledge the reviewer’s comment. However, most literature reviews that are conducted use title and abstract both for screening to identify potential interesting studies. Therefore, mentioning the name of circulatory miR-4798-3p only in our abstract seems to be sufficient to make sure our work will be identified, and recognized by others, if needed. In addition, we think the current title presents better our approach for a genome-wide miRNA profiling including hundreds of circulating miRNAs associated with AF.

2. The overall objective of this work is stated to be pathophysiology discovery. Finding an association of a biomarker to a disease is only step 1. Could this information be used for pharmaceutical targets? Is there a routine clinical laboratory role for measuring miR-4798-3p for early AF diagnosis, risk stratification, or therapeutic monitoring at least for males? The authors should speculate on the potential of these.
Response of the author:
As requested by the reviewer, we have added additional information in the Discussion where we speculate on other potential future implications “These future experimental studies could then further aid in early AF diagnosis, risk strati-fication, therapeutic monitoring or identification ofy potential interesting pharmaceutical drug targets among men.” on page 10, lines 340-343.

3. There is considerable amount of data from the literature review and in silico genetic analyses that are presented as supplementary tables. This reviewer suggests that this work could be a separate publication.
Response of the author:
We agree with the reviewer regarding the considerable amount of data from the literature review and in silico analyses. We believe that our manuscript is more comprehensive by presenting the literature review and in silico analyses in the supplementary tables, which gives an thorough overview about this topic to the readers.

4. The authors should identify which point on Figure 1 is the miR-4798-p3. Otherwise, this reviewer does not think this plot should be included in the main text.

Response of the author: We added this information directly beneath Figure 2 as a legend “The y-axis of Figure 12 represents the negative log of the p-value on the y-axis and the x-axis represents the log of the fold change for prevalent AF. The red dot indicates the significant association after correction for multiple testing of miR-4798-3p.” on page 8, lines 248-250.

Round 2

Reviewer 3 Report

The authors have complied with some of my suggestions but not others.